# LFRD: Enhancing Adversarial Transferability via Low-Rank Features Guidance and Representation Dispersion Regularization

## Abstract

Transfer-based adversarial attacks have become a mainstream approach for fooling modern deep neural networks. Numerous methods have aimed to enhance adversarial transferability by perturbing intermediate-layer features. However, existing methods overfit surrogate-specific features and generate imbalanced feature activations to unseen models. To address these issues, we propose LFRD, a transferable adversarial attack framework that combines low-rank features extraction and representation dispersion regularization. Specifically, Singular Value Decomposition (SVD) is employed to isolate low-rank components that capture dominant and invariant semantic features shared across models, providing model-free guidance and mitigating surrogate-specific overfitting. In parallel, a regularization term based on the Herfindahl–Hirschman Index (HHI) is introduced to balance feature activations by penalizing overly dominant responses and amplifying weaker ones. By jointly aligning perturbations with low-rank semantic structures and promoting dispersed feature utilization, LFRD yields adversarial examples with improved representation-level generalization. Experimental results on both standard and robust models show that our method demonstrates stronger adversarial transferability than state-of-the-art methods.

## 1 Introduction

Deep neural networks (DNNs) have achieved remarkable success in many computer vision tasks (Rawat & Wang, 2017; Li et al., 2018; Xu et al., 2019; Wang et al., 2021a; 2023), yet remain highly vulnerable to adversarial examples, subtle perturbations that mislead predictions while remaining imperceptible to humans. This vulnerability raises serious concerns for real-world applications (Deng et al., 2020; Hu et al., 2023; 2024) such as autonomous driving and face recognition. In black-box scenarios, where attackers lack access to the target model's parameters or architecture, the success of an attack depends on the transferability of adversarial examples crafted on source models. However, differences in model design and training often reduce transfer effectiveness. Thus, improving adversarial transferability is a critical and ongoing challenge in deep learning security.

Existing transfer-based attacks can be broadly categorized into two main approaches. The first class, including FGSM (Goodfellow et al., 2015), I-FGSM (Kurakin et al., 2018), MI-FGSM(Dong et al., 2018), and their variants (Dong et al., 2019; Lin et al., 2019; Xie et al., 2019; Jang et al., 2022; Zou et al., 2022), generates perturbations by optimizing gradients with respect to the output logits, effectively pushing the input toward the decision boundary of the source model. Although computationally efficient and easy to implement, these methods tend to overfit the source model's decision boundaries, leading to poor generalization on unseen architectures, particularly those with different inductive biases. In contrast, the second class seeks to improve transferability by perturbing intermediate-layer features, which often encode more stable and semantically meaningful representations across models. Among such approaches, Feature Importance-Aware Attack (FIA) (Wang et al., 2021b) emphasizes perturbing features based on their class-wise discriminative importance, encouraging alignment with task-relevant semantics. Neuron Attribution-Based Attack (NAA) (Zhang et al., 2022) extends this approach by employing integrated gradients to identify highly influential neurons, thereby refining the direction and focus of adversarial perturbation. Building on this direction, Intermediate-Level Perturbation Decay (ILPD) (Li et al., 2023) introduces

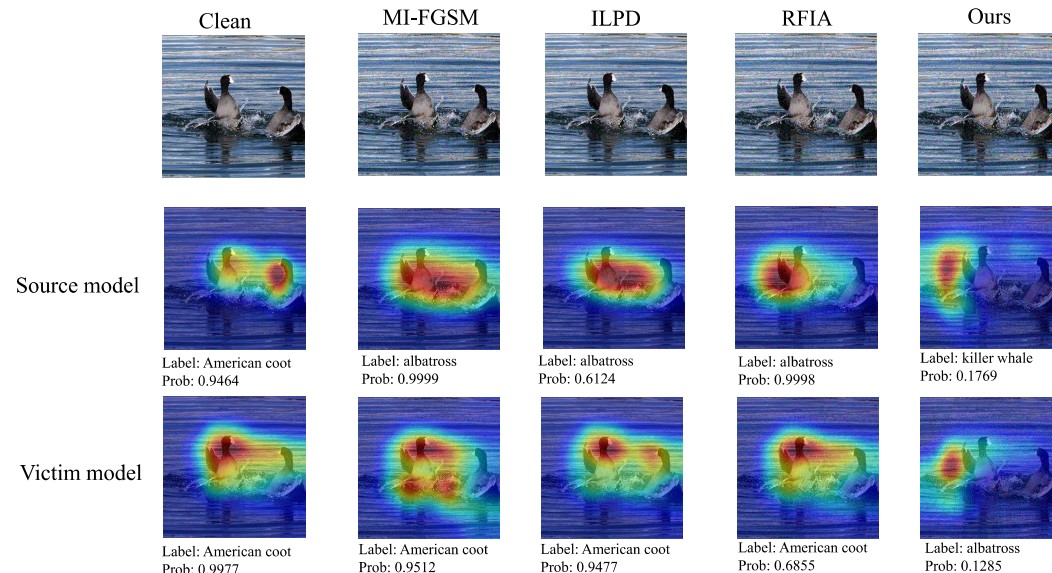

Figure 1: Visualization of attention shifts for clean images and adversarial examples generated by MI-FGSM, ILPD, RFIA and our method. Adversarial examples are crafted on the source model Inception-v3 and evaluated on the Victim model ResNet-50. Attention maps are computed using Eigen-CAM(Muhammad & Yeasin, 2020).

a decay mechanism that suppresses perturbation in less transferable channels. Prioritizing feature dimensions that better generalize across architectures. Recently, Relative Feature Importance-Aware Attack (RFIA) (Li et al., 2025) improves transferability by leveraging clean image gradients to construct relative feature importance, combining both dominant and noncritical semantics to guide perturbations that generalize across architectures. However, existing transfer-based methods often overfit surrogate-specific features and lack mechanisms to regulate perturbation effects on feature activations, leading to distortions dominated by a few high-activation units while neglecting low-activation regions. This imbalance narrows the coverage of perturbations and weakens generalization across different architectures.

In this paper, we propose LFRD, a transferable adversarial attack framework that integrates two complementary modules to enhance cross-model generalization. The first module employs Singular Value Decomposition (SVD) on intermediate features to isolate low-rank components that capture dominant semantic patterns shared across different architectures. These low-rank features provide model-free guidance and are fused with the original logits to form a multi-path optimization strategy. The second module introduces a representation dispersion regularization term based on the Herfindahl–Hirschman Index (HHI), which balances feature activations by suppressing overly dominant responses and amplifying weaker ones. This design prevents perturbations from being confined to a small set of high-activation regions and promotes broader utilization of perturbations. As shown in Figure 1, MI-FGSM, ILPD and RFIA primarily disrupt attention within the source model, but fail to sufficiently shift the victim model's focus, which still attends to the original semantic regions. In contrast, our method consistently redirects attention in both the source and victim models toward irrelevant regions, demonstrating stronger cross-model misdirection and improved transferability. In summary, our contributions are as follows:

- We propose LFRD, a transferable adversarial attack that improves transferability by reducing surrogate-specific overfitting and mitigating imbalanced feature activations.

- LFRD integrates two modules, including SVD-guided features extraction to identify architecture-invariant directions, and HHI-based regularization to encourage uniformly distributed feature activations across spatial and channel dimensions.

- Experiments on both normally and adversarially trained models demonstrate that LFRD, when combined with gradient stabilization, surpasses existing state-of-the-art transfer-based attacks in transferability performance.

## 2 RELATED WORK

**Adversarial Attacks and Transferability.** Adversarial attacks exploit the vulnerability of deep neural networks (DNNs) by introducing imperceptible perturbations that induce misclassification. In black-box settings, where model details are unavailable, transfer-based attacks generate adversarial examples on source models, requiring strong cross-model generalization. Early methods such as FGSM (Goodfellow et al., 2015), PGD (Madry et al., 2018), MI-FGSM (Dong et al., 2018), NI-FGSM (Lin et al., 2019), PI-FGSM (Gao et al., 2020), VMI-FGSM (Wang & He, 2021) and SVRE-MI (Xiong et al., 2022) optimize perturbations at the output layer but often overfit the source model's decision boundary, limiting transferability. To overcome this, intermediate-layer attacks like FDA (Ganeshan et al., 2019), NAA (Zhang et al., 2022), and FIA (Wang et al., 2021b) target more stable semantic representations shared across architectures. Extending this line of research, Relative Feature Importance-Aware Attack (RFIA) (Li et al., 2025) introduces a gradient-based strategy that constructs relative feature importance from clean-image activations, guiding perturbations along both dominant and non-dominant semantic dimensions.

**Singular Value Decomposition in Vision and Attack.** Singular Value Decomposition (Golub & Reinsch, 1971) is a classical tool in computer vision (Sadek, 2012), (Bermeitinger et al., 2019), (Levinson et al., 2020)that decomposes high-dimensional data into orthogonal components ranked by importance, enabling the extraction of dominant low-rank semantic patterns. Recent works have shown that the rank-1 component of intermediate features often encodes architecture-invariant semantics, making it a promising direction for enhancing adversarial transferability. Weng et al. Weng et al. (2024) introduced the use of SVD to guide perturbations along dominant directions via logit fusion.

**Dispersion Regularization and HHI.** In transferable adversarial attacks, ensuring broad spatial and semantic dispersion of perturbations is crucial for cross-model generalization, as overly concentrated noise often overfits surrogate-specific patterns. UFAF (Xu et al., 2024) proposes a dispersion loss and a distance loss to jointly guide transferable adversarial perturbations. Inspired by this, we introduce a regularization term based on the Herfindahl–Hirschman Index (HHI) regulate perturbation impact by suppressing overly dominant activations and enhancing weaker ones across spatial and channel dimensions. Originally used in economics to measure market concentration, HHI (Rhoades, 1993) reflects the degree to which activation energy is unevenly distributed.

## 3 METHODOLOGY

### 3.1 PRELIMINARY

We consider the standard setting of untargeted adversarial attacks in a black-box transfer scenario. Given an input image $x$ with ground-truth label $y$ and a source model $f$, the objective is to generate an adversarial example $x^{\text{adv}}$ such that

$$\operatorname{argmax} f'(x^{\text{adv}}) \neq y, \text{ and } \|x^{\text{adv}} - x\|_\infty \leq \epsilon \tag{1}$$

where $f'$ denotes the target (black-box) model inaccessible to the attacker, and $\epsilon > 0$ defines the allowed perturbation budget under the $l_\infty$-norm constraint. In this setting, adversarial examples are generated solely using the source model $f$, and evaluated for transferability by testing their success rate on unseen target models $f'$. Since output-layer decision boundaries often vary significantly across architectures, many transfer-based attacks operate not on the final logits, but on internal feature representations, which tend to encode more stable and semantically meaningful patterns.

### 3.2 LFRD FRAMEWORK OVERVIEW

To overcome the limited semantic generalization and spatial concentration observed in prior transfer attacks, we propose LFRD, which introduces two complementary modules to enhance adversarial

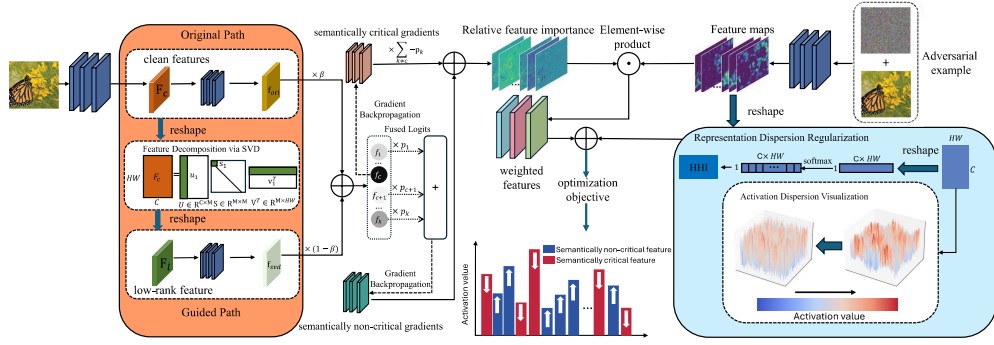

Figure 2: The overall framework of our proposed LFRD method. Based on the RFIA framework, it integrates a low-rank features guidance via SVD and a representation dispersion regularization module via HHI.

transferability. As illustrated in Figure 2, LFRD constructs a multi-path optimization framework by integrating a SVD-guided features path and a representation dispersion regularization. The SVD-guided feature path extracts a low-rank dominant direction from intermediate features via SVD, capturing transferable semantics shared across architectures. In parallel, an HHI-based dispersion regularizer penalizes peaked activation patterns by suppressing overly dominant responses and elevating weaker ones to disperse perturbation effects across spatial locations and channels. These modules steer perturbations toward model-free semantic features while avoiding concentration in a few regions.

### 3.3 LOW-RANK FEATURES GUIDANCE VIA SVD

To improve the semantic generalization capability of adversarial examples, LFRD introduces a secondary optimization path that explicitly models dominant transferable features using Singular Value Decomposition (SVD). The core idea is that the most semantically meaningful and architecture-invariant information within intermediate representations is often concentrated in a low-rank subspace. Instead of relying solely on gradients from the standard feature stream, which may contain model-specific or noisy activation patterns, this module extracts the principal semantic direction from the clean input and leverages it to guide adversarial optimization.

Given an intermediate feature map $h(x) \in \mathbb{R}^{C \times H \times W}$ from the source model, we reshape it into a 2D matrix $H \in \mathbb{R}^{C \times (H \times W)}$. We apply SVD to decompose it into:

$$H = U \Sigma V^T \tag{2}$$

We retain only the leading singular value $\sigma_1$ and its corresponding vectors $u_1 \in \mathbb{R}^C$, $v_1 \in \mathbb{R}^{H \times W}$, to reconstruct a rank-1 approximation:

$$Z = \sigma_1 u_1 v_1^T \tag{3}$$

This component is reshaped back to the original spatial dimensions to form a low-rank feature map $h^{\text{svd}} \in \mathbb{R}^{C \times H \times W}$, which serves as a parallel semantic representation of the clean input. To integrate this semantic guidance into the attack process, we compute logits from both the feature $h(x)$ and the low-rank feature $h^{\text{svd}}$, denoted as $f_{\text{ori}}$ and $f_{\text{svd}}$, respectively. These logits are linearly fused using a hyperparameter $\beta \in [0, 1]$:

$$f_{\text{fused}} = \beta f_{\text{ori}} + (1 - \beta) f_{\text{svd}} \tag{4}$$

Finally, the classification loss is computed on the fused logits:

$$\mathcal{L}_{\text{cls}} = \mathcal{L}_{\text{CE}}(f_{\text{fused}}, y) \tag{5}$$

This design enables dual-path gradient optimization: one that captures fine-grained discriminative patterns from the original feature maps, and another that follows the global, architecture-agnostic semantic direction derived via SVD.

## 3.4 REPRESENTATION DISPERSION REGULARIZATION VIA HHI

While semantic guidance steers perturbations toward transferable features, we observe that the resulting distortions can become concentrated in a few spatial locations or channels, dominating limited regions of the feature space. Such concentration aligns with surrogate-specific attention patterns and undermines generalization to architectures with different inductive biases. To mitigate this issue, LFRD introduces a representation dispersion regularization based on the Herfindahl–Hirschman Index (HHI).

In the context of adversarial learning, we reinterpret HHI as a differentiable indicator of how activation energy is distributed across a feature tensor. Given an intermediate activation map $h(x^{\text{adv}}) \in \mathbb{R}^{C \times H \times W}$, we treat its absolute values as an unnormalized energy distribution over all spatial and channel dimensions. We first flatten and normalize the tensor into a probability vector:

$$E = \text{softmax}\left(\left|h(x^{\text{adv}})\right|\right) \in \mathbb{R}^{C \times H \times W} \tag{6}$$

Each row of $E$ represents a normalized energy distribution over the feature dimensions for a single image.

The Herfindahl-Hirschman Index is then computed for each sample as:

$$HHI(E) = \sum_{i=1}^{D} E_i^2 \tag{7}$$

Higher HHI indicates a peaked (dominated) activation distribution, whereas lower HHI indicates a more dispersed influence across elements. We therefore define the dispersion loss as the mean HHI and minimize it:

$$\mathcal{L}_{\text{HHI}} = 1 - (HHI(E)) \tag{8}$$

Minimizing $\mathcal{L}_{\text{HHI}}$ penalizes overly dominant activations and relatively elevates weaker ones, dispersing the perturbation's effect over spatial and channel dimensions and disrupting feature representations more broadly.

Finally, LFRD integrates the classification loss and the representation dispersion regularization into a unified objective:

$$\mathcal{L}_{\text{LFRD}} = \lambda \mathcal{L}_{\text{cls}} + \mathcal{L}_{\text{HHI}} \tag{9}$$

where $\lambda$ is a tunable coefficient that controls the relative strength of semantic guidance in the optimization. The dispersion term is assigned a fixed weight to consistently enforce perturbation spread across feature space. On top of MI-FGSM, we generate the final adversarial examples by iteratively ascending the gradient of $\mathcal{L}_{\text{LFRD}}$.

## 4 EXPERIMENTS

### 4.1 EXPERIMENTAL SETTING

**Dataset.** Following recent works, we construct the attack dataset by randomly sampling 1000 images from the ILSVRC 2012 validation set (Russakovsky et al., 2015), ensuring that each image is correctly classified by all target models. This design aligns with prior studies and enables a fair comparison with state-of-the-art transfer-based adversarial attacks.

**Models.** For generating adversarial examples, we select five commonly used source models with diverse architectures: ResNet-50(Res-50) (He et al., 2016), ResNet-152 (Res-152) (He et al., 2016), Vgg19 (Simonyan & Zisserman, 2014) , Inception-v3(Inc-v3) (Szegedy et al., 2016), and Inception-v4(Inc-v4) (Szegedy et al., 2017). To evaluate the black-box transferability, we test the generated adversarial examples on both normally trained and adversarially trained models:

**Normally trained models:** ResNet-50, ResNet-152, DenseNet-121 (Huang et al., 2017), Inception-v3, Inception-v4, Vgg19, MLP-Mixer-b (Tolstikhin et al., 2021), ConvNeXt-T (Liu et al., 2022), ViT-B (Dosovitskiy et al., 2020), DeiT-B (Touvron et al., 2021), and Swin-B (Liu et al., 2021).

Table 1: Attack success rate (%) on 11 normally trained models for different attack methods.

| Source model | Methods | Res-50 | Res-152 | Inc-v3 | Inc-v4 | Vgg-19 | Mixer-b | Conv next-T | ViT-b | DeiT-b | Swin-b | Dense-121 | Avg |
|---|---|---|---|---|---|---|---|---|---|---|---|---|---|
| Res-50 | MIM | 100 | 88.7 | 64.9 | 64.8 | 75.4 | 46.0 | 36.9 | 27.2 | 28.0 | 20.9 | 84.8 | 57.96 |
| | VMI | 99.9 | **98.3** | 86.8 | 83.4 | 88.6 | 64.7 | 62.9 | 46.3 | 48.9 | 39.5 | 97.0 | 74.21 |
| | ILPD | 99.8 | 94.9 | 75.2 | 76.5 | 82.7 | 49.5 | 56.0 | 30.8 | 33.5 | 32.9 | 90.6 | 65.67 |
| | RFIA-AB | 100 | 97.4 | 89.0 | 88.1 | 90.8 | 66.5 | 70.0 | 49.9 | **52.1** | 47.0 | 96.4 | 77.01 |
| | LFRD-AB | **100** | 97.3 | **92.2** | **91.6** | **93.4** | **69.2** | **71.9** | **52.8** | 51.7 | **48.8** | **97.3** | **78.74** |
| Res-152 | MIM | 92.7 | 100 | 65.2 | 63.4 | 74.3 | 46.1 | 39.8 | 27.6 | 30.2 | 23.0 | 85.7 | 58.90 |
| | VMI | **98.4** | 100 | 86.2 | 82.6 | 86.2 | 64.8 | 69.8 | 50.2 | 54.9 | 45.5 | 96.8 | 75.94 |
| | ILPD | 95.2 | 99.9 | 74.7 | 74.5 | 76.7 | 49.5 | 57.3 | 32.0 | 37.4 | 37.1 | 91.4 | 65.97 |
| | RFIA-AB | 97.7 | 100 | 87.7 | 84.6 | 85.2 | 67.3 | 68.0 | 52.2 | 53.3 | 47.4 | 94.7 | 76.19 |
| | LFRD-AB | 97.7 | **100** | **91.3** | **89.5** | **88.6** | **70.9** | **72.1** | **55.7** | **56.4** | **51.7** | **96.8** | **79.15** |
| Inc-v3 | MIM | 49.6 | 42.5 | 99.4 | 60.7 | 65.4 | 36.6 | 20.8 | 20.2 | 17.6 | 10.5 | 45.6 | 42.62 |
| | VMI | 68.8 | 63.7 | 99.6 | 75.3 | 74.6 | 48.0 | 38.7 | 31.0 | 30.2 | 24.6 | 64.9 | 56.30 |
| | ILPD | 52.8 | 49.7 | 95.8 | 65.6 | 63.5 | 33.3 | 30.5 | 18.8 | 19.3 | 17.7 | 49.9 | 45.17 |
| | RFIA-ABC | 78.9 | 75.2 | **99.8** | 85.2 | 81.9 | 54.9 | 52.8 | **36.8** | 32.7 | 30.9 | 76.8 | 64.17 |
| | LFRD-ABC | **80.1** | **76.6** | 99.0 | **86.9** | **83.3** | **55.8** | **53.7** | 36.6 | **33.2** | **32.3** | **79.0** | **65.13** |
| Inc-v4 | MIM | 45.9 | 38.8 | 62.1 | 97.7 | 64.8 | 36.2 | 27.3 | 19.4 | 17.2 | 14.3 | 41.4 | 42.28 |
| | VMI | 65.9 | 61.4 | 77.8 | 97.9 | 75.3 | 47.4 | 48.6 | 31.9 | 33.2 | 29.2 | 62.0 | 57.32 |
| | ILPD | 43.7 | 42.1 | 57.6 | 90.7 | 63.5 | 32.7 | 35.4 | 18.8 | 19.8 | 20.3 | 41.9 | 42.40 |
| | RFIA-ABC | 74.0 | 69.1 | 83.2 | **99.1** | 80.1 | 52.6 | 55.4 | **35.4** | **33.7** | **34.6** | 69.5 | 62.42 |
| | LFRD-ABC | **74.8** | **70.2** | **84.1** | 98.6 | **81.2** | **53.7** | **57.4** | 34.9 | 32.7 | 34.1 | **72.1** | **63.07** |
| Vgg19 | MIM | 74.4 | 60.8 | 67.8 | 73.1 | 99.8 | 44.2 | 44.6 | 23.4 | 24.0 | 20.8 | 68.2 | 54.64 |
| | VMI | 89.3 | 78.3 | 81.1 | 85.7 | 100 | 55.9 | 62.7 | 36.3 | 37.2 | 34.6 | 81.7 | 67.52 |
| | ILPD | 86.6 | 76.0 | 76.9 | 84.7 | 99.9 | 44.6 | 67.6 | 25.1 | 25.1 | 33.9 | 80.3 | 63.70 |
| | RFIA-AB | 91.5 | 85.1 | 89.7 | 93.9 | 100 | 61.8 | 74.6 | 41.6 | 40.8 | **46.0** | 89.6 | 74.05 |
| | LFRD-AB | **92.9** | **86.5** | **91.2** | **95.2** | 100 | 61.8 | **76.0** | **42.0** | **41.4** | 44.4 | **90.8** | **74.74** |

**Adversarially trained robust models:** Inc-v3$_{adv}$ (Madry et al., 2017), Inc-v3$_{ens3}$, Inc-v3$_{ens4}$, IncRes-v2$_{adv}$, IncRes-v2$_{ens3}$ (Tramèr et al., 2017), EfficientNet-b0(robust), NFNet-l0(robust), PVT-v2-b0(robust), and Sequencer2d-s(robust).

**Baseline Methods.** We choose MI-FGSM(MIM) as the basic gradient-based baseline, and include several advanced variant such as VMI. To highlight the advantage of semantic-guided feature perturbation, we also compare LFRD with leading intermediate-level perturbation methods , including FIA, NAA, ILPD, and RFIA.

**Parameters.** Following the experimental setup in prior works, we adopt consistent parameter settings to ensure fair comparison and reproducibility. The maximum perturbation$\epsilon$ is set to $16/255$, with an iteration step size of $1.6/255$ and a total of 10 iterations across all attacks. Momentum is applied uniformly to stabilize updates, using a decay factor of $\mu = 1.0$. VMI uses $N = 10$ gradient samples per iteration, with the neighborhood radius bounded at $\beta = 1.5 \times \epsilon$. For all intermediate-layer-based methods including FIA, NAA, ILPD, RFIA and our LFRD which we select specific layers from each source model as target feature blocks. Specifically, we use layer2.4 for ResNet-50, layer3.6 for ResNet-152, Mixed-6b for Inception-v3, Reduction-A for Inception-v4, and Conv3-4 for Vgg-19. In methods requiring gradient aggregation , Gaussian noise with zero mean and a variance of $0.1$ is added to the input at each iteration to promote robustness. The dropout probability in FIA is fixed at $0.1$. Finally, the number of aggregations in FIA, NAA, ILPD, RFIA, and our LFRD method is uniformly set to 10, ensuring equal computational complexity across all approaches compared. In $f_{\text{fused}}$, the hyperparameter $\beta$ is set to 0.5.

**Robust Gradient Stabilization Strategies.** To ensure stable and transferable gradient signals during adversarial optimization, we follow RFIA and apply three widely used robust gradient strategies: Integrated Gradient (IG) (Sundararajan et al., 2017), SmoothGrad (SG) (Smilkov et al., 2017), and

Table 2: Attack success rate (%) on 9 adversarially trained models for different attack methods.

| Source model | Methods | Adv-Inc-v3 | Adv-Incres-v2 | Ens3-Adv Inc-v3 | Ens4-Adv Inc-v3 | Ens-Incres-v2 | Efficientnet-b0 | Nfnet-10 | pvt-v2 | Sequencer-2d | Avg |
|---|---|---|---|---|---|---|---|---|---|---|---|
| Res-50 | MIM | 42.4 | 37.2 | 41.7 | 38.6 | 29.8 | 66.6 | 46.7 | 66.2 | 37.0 | 45.13 |
| | VMI | 73.5 | 69.7 | 72.2 | 70.0 | 61.6 | 90.3 | 72.4 | 85.9 | 64.5 | 73.34 |
| | ILPD | 59.1 | 50.4 | 56.6 | 53.1 | 40.7 | 82.2 | 63.6 | 79.0 | 56.3 | 60.11 |
| | RFIA-AB | 77.4 | 73.3 | 75.8 | 74.6 | 65,1 | 92.4 | 76.7 | 88.0 | 69.0 | 76.91 |
| | LFRD-AB | **82.1** | **77.5** | **78.6** | **75.1** | **68.2** | **92.4** | **78.2** | **88.2** | **70.6** | **79.21** |
| Res-152 | MIM | 45.3 | 43.5 | 43.4 | 43.7 | 32.6 | 69.7 | 45.8 | 63.4 | 40.4 | 47.53 |
| | VMI | 78.8 | 76.6 | 77.8 | **76.4** | **70.1** | 89.3 | 76.3 | 84.8 | 68.7 | 77.65 |
| | ILPD | 62.2 | 56.1 | 58.8 | 54.5 | 45.1 | 78.6 | 66.0 | 75.0 | 57.5 | 61.54 |
| | RFIA-AB | 77.0 | 75.4 | 74.3 | 72.1 | 63.6 | 88.4 | 74.2 | 83.6 | 67.4 | 75.11 |
| | LFRD-AB | **83.4** | **81.2** | **78.9** | 75.8 | 69.5 | **93.0** | **78.0** | **87.4** | **71.1** | **79.81** |
| Inc-v3 | MIM | 26.7 | 23.9 | 22.4 | 21.9 | 10.1 | 44.9 | 26.8 | 46.2 | 21.8 | 27.66 |
| | VMI | 45.5 | 45.1 | 42.2 | 42.6 | 24.6 | 66.2 | 46.4 | 62.5 | 40.6 | 46.74 |
| | ILPD | 30.6 | 30.4 | 29.2 | 30.3 | 16.8 | 49.5 | 35.9 | 51.5 | 28.1 | 33.59 |
| | RFIA-ABC | 54.2 | 56.8 | **43.1** | **42.8** | **25.3** | 78.6 | 58.4 | 74.6 | 50.0 | 53.76 |
| | LFRD-ABC | **54.6** | **57.6** | 42.3 | 42.0 | 25.2 | **79.6** | **59.0** | **76.2** | **50.7** | **54.13** |
| Inc-v4 | MIM | 23.1 | 21.3 | 18.1 | 18.2 | 11.4 | 46.6 | 32.1 | 43.1 | 22.7 | 26.29 |
| | VMI | 42.8 | 44.4 | 41.2 | 40.6 | **27.7** | 65.8 | 55.3 | 61.4 | 44.7 | 47.65 |
| | ILPD | 24.0 | 26.2 | 26.3 | 26.4 | 16.3 | 43.9 | 41.5 | 44.2 | 31.4 | 31.13 |
| | RFIA-ABC | 46.8 | 50.6 | 43.0 | **41.3** | 24.7 | 74.0 | 63.6 | 68.7 | 51.1 | 51.54 |
| | LFRD-ABC | **48.1** | **53.9** | **44.1** | 40.9 | 25.5 | **75.1** | **65.6** | **71.1** | **52.4** | **52.96** |
| Vgg19 | MIM | 38.7 | 32.5 | 35.4 | 33.8 | 23.0 | 71.1 | 54.5 | 73.3 | 41.7 | 44.89 |
| | VMI | 60.9 | 53.4 | 56.7 | 54.3 | 40.7 | 83.6 | 72.1 | 87.4 | 59.6 | 63.41 |
| | ILPD | 53.2 | 38.3 | 46.2 | 41.7 | 27.7 | 82.6 | 74.1 | 83.4 | 60.2 | 56.37 |
| | RFIA-AB | 72.6 | 64.9 | 67.8 | 62.7 | 51.0 | 90.6 | **82.1** | 90.3 | **70.4** | 72.60 |
| | LFRD-AB | **75.7** | **65.3** | **69.2** | **63.3** | **51.3** | **91.4** | 81.6 | **90.9** | 69.5 | **73.13** |

Gradient Accumulation (GA) (Wang et al., 2021b), which respectively aim to alleviate gradient saturation, suppress noisy gradients, and reduce model-specific information. These strategies enhance the computation of relative feature importance and consistently improve cross-model transferability. For clarity in comparison, we append suffixes to each attack method to indicate the stabilization strategies used: -A corresponds to IG, -B to SG, and -C to GA, with -AB, -AC, -BC, and -ABC denoting their respective combinations. For example, LFRD-ABC refers to the variant where all three strategies are applied, while LFRD-B indicates that only SmoothGrad is used.

## 4.2 ATTACK RESULTS

In this section, we evaluate the proposed LFRD framework across a comprehensive set of normally and adversarially trained models to validate its effectiveness in enhancing adversarial transferability. It is necessary to explain how the three robust strategies ought to be applied when integrated with LFRD (for instance, LFRD-ABC). Actually, no special attention is paid to the order in which these strategies are applied. For example, regarding LFRD-ABC, strategies A, B, and C are merely implemented in sequence.

We compare our proposed method (LFRD) with MIM and its variant (VMI), as well as with ILPD and RFIA, as shown in Table 1 and Table 2, which present the average attack success rates across 11 normally trained models and 9 adversarially trained models. It can be clearly observed that our method consistently outperforms the others in most cases, and the average results are always superior. The - notation indicates the best-performing combination of robust gradient stabilization strategies for each method on each source model. It can be observed that the optimal strategy selection for RFIA and our LFRD method remains consistent across different source models. Specifically, the ABC combination demonstrates better adaptability on Inc-v3 and Inc-v4, while the AB combination proves to be more effective on Res-50, Res-152, and Vgg19. These results suggest that selecting appropriate gradient stabilization strategies tailored to the characteristics of each source model is crucial for enhancing attack transferability.

Table 3: Attack success rates (%) of ILPD, FIA, NAA, RFIA and our LFRD with different combinations (A, B, C, AB, AC, BC, ABC), averaged on 11 normally trained models (□) and 9 adversarially trained models (△). Adversarial examples are crafted using Inc-v3 as the source model.

| Gradient Strategies | Target Models | ILPD | FIA | NAA | RFIA | LFRD |
|---|---|---|---|---|---|---|
| A | □ | 52.20 | 51.65 | 56.59 | 58.14 | **60.33** |
|   | △ | 40.62 | 40.02 | 45.32 | 47.66 | **48.26** |
| B | □ | 57.83 | 53.68 | 57.36 | 58.72 | **59.89** |
|   | △ | 44.70 | 44.68 | 48.86 | 49.31 | **50.80** |
| C | □ | 58.44 | 57.20 | 60.59 | 61.58 | **62.38** |
|   | △ | 47.66 | 47.16 | 50.22 | 50.75 | **51.66** |
| AB | □ | 58.93 | 56.85 | 60.51 | 61.54 | **62.77** |
|   | △ | 47.65 | 46.92 | 49.72 | 51.36 | **52.44** |
| AC | □ | 57.68 | 60.19 | 62.55 | 63.75 | **64.76** |
|   | △ | 47.96 | 49.38 | 51.46 | 52.98 | **53.68** |
| BC | □ | 57.81 | 59.71 | 62.53 | 63.39 | **63.78** |
|   | △ | 48.52 | 49.11 | 52.11 | 52.98 | **53.86** |
| ABC | □ | 59.58 | 61.28 | 63.03 | 64.89 | **65.13** |
|   | △ | 48.52 | 50.90 | 52.47 | 53.43 | **54.13** |

To eliminate the influence of robust gradient stabilization strategies and their combinations on the experimental results, we conducted a comprehensive evaluation of ILPD, FIA, NAA, RFIA and our LFRD by integrating them with the three individual strategies (A, B, C) and all possible combinations (AB, AC, BC, ABC). The corresponding attack success rates averaged on 11 normally trained models and 9 adversarially trained models are reported in Table 3. The results show that, regardless of the specific strategy or combination used, our LFRD consistently achieves higher average attack success rates than the other baseline methods, demonstrating superior transferability.

Extensive experiments demonstrate that the proposed LFRD framework achieves superior adversarial transferability across both standard and adversarially trained models. This performance stems from the integration of SVD-guided low-rank features and HHI-based dispersion, which together steer perturbations toward model-free shared features and disperse their impact across spatial and channel dimensions by suppressing overly dominant activations and elevating weaker ones. Combined with robust gradient strategies, these components enhance optimization stability and cross-model generalization, making LFRD a highly effective approach for transferable adversarial attacks.

## 4.3 ABLATION STUDY

**Component Effectiveness.** To further investigate the effectiveness of each proposed component in LFRD, we conduct an ablation study using three representative source models: ResNet-50, Inception-v4, and Vgg19. As shown in Table 4, adding either the SVD-guided features path or the HHI-based representation dispersion regularization to the baseline (RFIA) results in a consistent improvement in attack success rates. Specifically, applying only SVD enhances features guidance and yields notable gains on Vgg19, while using HHI alone promotes spatial and channel dispersion, leading to higher performance on ResNet-50. When both components are combined in the full LFRD framework, a cumulative effect is observed, achieving the best results across all source models. These findings confirm the complementary strengths of the two modules in boosting adversarial

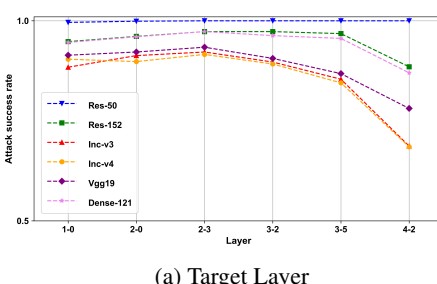
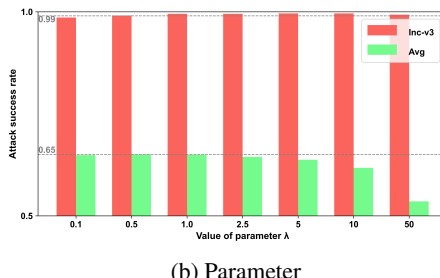

(a) Target Layer                        (b) Parameter

Figure 3: Ablation study on target layer and parameter. (a) Attack success rates using different target layers. Adversarial examples are crafted using Res-50 as the source model. (b) Attack success rates on Inc-v3 (red) and attack success rates averaged on 11 normally trained models (green) under different $\lambda$ values. Adversarial examples are crafted using Inc-v3 as the source model.

Table 4: Ablation results of low-rank features guidance (SVD) and representation dispersion regularization (HHI) on three source models. Results are Attack success rate (%) averaged on 11 normally trained models.

| Methods | Res-50 | Inc-v4 | Vgg19 |
|---|---|---|---|
| Baseline (RFIA) | 78.02 | 61.94 | 72.84 |
| +SVD only | 78.26 | 62.76 | 74.67 |
| +HHI only | 78.55 | 62.82 | 73.51 |
| LFRD (Ours) | **78.74** | **63.07** | **74.74** |

transferability. During evaluation, we adopt the best-performing gradient stabilization configuration (AB or ABC) for each variant to ensure fair and consistent comparison.

**Target Layer.** Considering that the choice of target layer can affect the adversarial examples generated by our method, it is necessary to determine which layer of the source model yields the most effective attacks. As shown in the Figure 3a , selecting mid-level layers consistently results in higher attack success rates compared to shallow or deep layers. This suggests that mid-level features exhibit greater semantic consistency across different architectures, while both shallow and deep layers contain more model-specific information.

**Parameter.** To evaluate the sensitivity of LFRD to the weighting factor $\lambda$, we conduct experiments by varying $\lambda$ in the Equation 9. As shown in Figure 3b, setting $\lambda = 1.0$ achieves a better balance between semantic guidance and representation dispersion regularization. A too-small $\lambda$ underutilizes semantic supervision and unsaturated white-box performance, while a too-large value weakens dispersion and reduce transferability. This demonstrates the necessity of jointly optimizing both components for robust transferability.

# 5 CONCLUSION

In this paper, we propose LFRD, a novel adversarial attack method that enhances transferability by integrating SVD-based low-rank features guidance and HHI-based representation dispersion regularization. These components together steer perturbations toward model-shared features while dispersing activation influence across space and channels. LFRD yields adversarial examples with improved representation-level generalization. Extensive experiments on both standard and robust models confirm that LFRD outperforms existing transfer-based attacks, demonstrating strong generalization and stability across architectures. In the future work, we will focus on multi-layer selection and alternative dispersion penalties beyond HHI to further improve transferability.

ETHICS STATEMENT

This research investigates transfer-based adversarial attacks to better understand and benchmark the robustness of deep neural networks. No human subjects or animal experiments were involved. We use publicly available datasets (ILSVRC 2012 ImageNet validation set) under their licenses and do not process personally identifiable information. We acknowledge the dual-use nature of adversarial attack. All experiments are conducted for scientific evaluation, and our releases are intended to support robustness research. We will refrain from deploying or encouraging use in real-world systems and will follow community norms for responsible disclosure and dissemination.

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

APPENDIX

## A  LLM USAGE

We used a Large Language Model only for language polishing. It did not contribute to ideas, methods, analyses, experiments, or results. All scientific content is the authors' own. We take full responsibility for the manuscript and ensured any LLM-edited text follows ethical guidelines and avoids plagiarism or misconduct.

