# OpenReview forum: "LFRD: Enhancing Adversarial Transferability via Low-Rank Features Guidance and Representation Dispersion Regularization"
_ICLR.cc/2026/Conference — ICLR 2026 Conference Withdrawn Submission_

### Official Review · Reviewer_S1iq · 2025-10-19

**Soundness:** 2
**Presentation:** 3
**Contribution:** 2
**Rating:** 4
**Confidence:** 3

**Summary:**

This paper introduces LFRD, a transfer-based adversarial attack framework designed to improve adversarial transferability by addressing two common limitations of existing feature-level attacks: overfitting to surrogate-specific features and unbalanced feature activations. The method integrates two key modules: (1) Low-Rank Feature Guidance via Singular Value Decomposition (SVD), which isolates dominant low-rank components to capture architecture-invariant semantics, and (2) Representation Dispersion Regularization using the Herfindahl–Hirschman Index (HHI) to penalize overly concentrated activations and promote balanced feature utilization. Extensive experiments on both normally trained and adversarially trained models demonstrate consistent improvements in transferability compared to state-of-the-art methods such as RFIA and ILPD. Ablation studies confirm that both modules contribute complementary benefits, and parameter analysis further validates robustness across different architectures.

**Strengths:**

1. The methodology is explained clearly, mathematical formulations are well presented, and implementation details (e.g., layer choices, parameter settings, gradient stabilization strategies) are sufficient to reproduce results.

2. Improving transfer-based attacks is an important topic for robustness research, and this paper’s ideas could inspire further exploration of low-rank structures and dispersion metrics in adversarial learning.

**Weaknesses:**

1.  While the intuition behind using SVD and HHI is sound, the paper lacks a deeper theoretical justification or analysis connecting low-rank structures and transferability beyond empirical observation. A discussion of why these components generalize better across architectures (e.g., via invariance theory or spectral analysis) would strengthen the contribution.

2. The method builds directly upon RFIA and shares much of its training pipeline. As a result, some novelty resides more in the combination of modules rather than a fundamentally new attack paradigm.

3. The introduction of SVD per iteration could be computationally heavy, especially for large feature maps, but the paper does not report training or inference overhead. This omission leaves open questions about efficiency and scalability.

4. Although generally clear, the writing occasionally suffers from redundancy and inconsistent notation (e.g., inconsistent subscript formatting and references to different stabilization strategies). Tightening exposition would improve readability.

5. The evaluation is restricted to ImageNet-based architectures. It would be interesting to see whether the proposed framework generalizes to other modalities or tasks, such as NLP or multimodal models, where feature dispersion might behave differently.

**Questions:**

Please refer to Weaknesses.

---

### Official Review · Reviewer_Biwr · 2025-10-31

**Soundness:** 2
**Presentation:** 3
**Contribution:** 2
**Rating:** 4
**Confidence:** 4

**Summary:**

This paper proposes LFRD, a new transfer-based adversarial attack framework designed to enhance adversarial transferability by addressing two key issues: overfitting to surrogate-specific features and imbalanced feature activations. LFRD's main contribution is its novel integration of two complementary modules: 1) Low-rank Features Guidance, which uses Singular Value Decomposition (SVD) to extract dominant, model-invariant semantic features to guide the perturbation and mitigate overfitting, and 2) Representation Dispersion Regularization, which employs the Herfindahl-Hirschman Index (HHI) to penalize overly concentrated feature responses and promote a more dispersed, balanced activation spread. By jointly optimizing these two components, LFRD generates adversarial examples with improved representation-level generalization, demonstrating stronger attack success rates against both standard and adversarially trained models compared to state-of-the-art methods.

**Strengths:**

1) Novel module combination: The integration of SVD-guided low-rank feature extraction and HHI-based dispersion regularization addresses two core limitations of existing methods in a complementary manner, with ablation studies confirming the value of each component.

2) Comprehensive experimental validation: The paper tests LFRD across diverse model architectures (CNNs, transformers) and training paradigms (normal, adversarial), ensuring the method’s generalizability.

3) Clear motivation and problem framing: The authors effectively articulate the issues of surrogate-specific overfitting and imbalanced activations, making the need for LFRD’s dual-module design intuitive.

**Weaknesses:**

1) Lack of theoretical depth：The paper's central hypothesis, stated as "The core idea is that the most semantically meaningful and architecture invariant information within intermediate representations is often concentrated in a low-rank subspace", is a strong assumption that lacks sufficient theoretical analysis or direct proof. This lack of justification is a significant weakness. Furthermore, the empirical evidence provided in the ablation study (Table 4) is not compelling enough to validate this claim. The performance gain from adding "+SVD only" is marginal at best (e.g., 78.02% to 78.26% for Res-50; 61.94% to 62.76% for Inc-v4), which raises doubts about the practical significance of this component and the core assumption it is built upon.

2) Hyperparameter under-exploration：The logit fusion weight $\beta$ in Equation (4) is set to 0.5 without clear justification. It is unclear how sensitive the attack performance of LFRD is to this specific value. A sensitivity analysis or ablation study on the hyperparameter $\beta$ is needed to validate this setting and understand its impact on the method's effectiveness.

3) Limited novelty over recent work: The relationship to Weng et al. (2024) is insufficiently distinguished. Both use SVD for extracting transferable directions. The paper must clarify the conceptual and technical differences more explicitly.

**Questions:**

1) Relationship to Weng et al. (2024): Can you explicitly compare your SVD approach with theirs? What are the key technical differences, and can you provide an empirical comparison?
2) Theoretical justification: Can you provide evidence (empirical or theoretical) that low-rank components of intermediate features are indeed more similar across different architectures? For instance, measuring subspace alignment or canonical correlation?
3) Computational cost: SVD at each iteration adds significant computational cost (O(C²HW) for standard algorithms). What is the wall-clock time overhead of LFRD compared to baselines? How does SVD scale with feature dimensions?
4) Statistical significance: Can you provide confidence intervals or significance tests for the improvements over RFIA, especially where gains are marginal (1-2%)?

---

### Official Review · Reviewer_hWP1 · 2025-10-31

**Soundness:** 3
**Presentation:** 3
**Contribution:** 2
**Rating:** 4
**Confidence:** 4

**Summary:**

This paper proposes LFRD, a transfer-based adversarial attack framework that aims to improve black-box transferability by (i) guiding perturbations with low-rank semantic features extracted via SVD from intermediate representations, and (ii) promoting representation dispersion with a regularizer based on the Herfindahl–Hirschman Index (HHI). Extensive experiments demonstrate that the proposed LFRD suppressed baseline method over diverse source models.

**Strengths:**

1. Clearly state the motivation that existing intermediate-level attacks can overfit surrogate features and concentrate perturbations; LFRD directly targets these two issues.
2 . The SVD and HHI are model-agnostic addition which plug into RFIA-style pipelines with minimal changes, showing good generality
3. The empirical coverage is wide, including diverse surrogate and target models. Also provide ablation studies to demonstrate the effectiveness of these two modules.

**Weaknesses:**

1. The paper states (line 193) that “most semantically meaningful and architecture-invariant information within intermediate representations is often concentrated in a low-rank subspace”, yet provides neither citations nor experiments to support this claim. Additional evidence showing that low-rank feature guidance indeed captures semantic, architecture-invariant patterns would make the motivation more convincing.
2. While the HHI term is introduced to promote representation dispersion, the paper lacks visualization or quantitative analysis demonstrating this effect. Without activation heatmaps or dispersion metrics, it remains unclear whether HHI meaningfully diversifies the activation distribution.
3. The method is only tested against standard and adversarially trained models. The absence of results on current defense mechanisms (e.g., DiffPure or other purification-based defenses) limits understanding of LFRD’s robustness and transferability under realistic defense conditions.
4. All reported results use CNNs as source models. Evaluating LFRD with ViTs as sources (not only as targets) would strengthen the claim that low-rank feature guidance promotes architecture-invariant semantics and general cross-architecture transferability.

**Questions:**

1. Please provide quantitative metrics or visual evidence (such as activation heatmaps) to demonstrate that the proposed dispersion regularization indeed produces more evenly distributed activations across spatial or channel dimensions.
2. The claim in line 193 requires empirical or cited evidence. Please provide supporting references or experiments demonstrating that low-rank feature guidance effectively captures semantic, architecture-agnostic components that enhance transferability.
3. Why is the low-rank guidance restricted to rank-1? Have you evaluated higher ranks (e.g., rank-2, rank-4) to examine whether increasing the subspace dimension further improves or degrades transferability? Please include ablation results and discuss how the optimal rank may vary with source architecture depth or feature dimensionality.
4. It would strengthen the paper to include attack results against defense-enhanced to further validate the robustness and superiority of LFRD under more realistic, defense-aware conditions.

---

### Official Review · Reviewer_J5Qc · 2025-11-05

**Soundness:** 3
**Presentation:** 2
**Contribution:** 2
**Rating:** 4
**Confidence:** 4

**Summary:**

The paper proposes LFRD, a transfer-based adversarial attack method that aims to enhance cross-model transferability by guiding perturbations with low-rank feature representations and introducing a representation dispersion regularizer. Specifically, the method applies Singular Value Decomposition (SVD) to extract “semantically critical” low-rank components from intermediate network features and uses them to guide the direction of perturbation updates. In addition, it introduces a Herfindahl–Hirschman Index (HHI)-based dispersion loss to prevent over-concentrated activations, thereby increasing feature diversity and transferability. Experiments are conducted on a single benchmark dataset using several normally trained and adversarially trained models to demonstrate the method’s effectiveness compared to prior transfer attack methods such as MIM, VMI, and RFIA.

**Strengths:**

1. Effort to explore architectural invariance.
By using low-rank guidance, the authors implicitly attempt to capture model-agnostic semantic information, which is a meaningful step toward transferable adversarial examples across heterogeneous architectures.

2. Empirical improvements over some feature-based baselines.
The reported results show modest but consistent gains on several victim models, suggesting that the proposed modules have some practical effect.

**Weaknesses:**

1. Unclear conceptual explanation in Fig. 2.
The meaning of “semantic critical” and “non-critical” gradients is ambiguous. While gradients can be computed from the loss, the distinction between “semantically critical” and “non-critical” components is not well justified. The figure gives the impression of two types of gradients without defining how they are obtained or how they relate to semantics. A formal mathematical definition or visualization of semantic attribution would help clarify this concept

2. Inconsistent problem setup in Fig. 2.
The figure already shows adversarial examples as the input to the framework, but the method’s stated goal is to generate adversarial examples with higher transferability. This raises confusion about where the initial adversarial examples come from and how the proposed process improves transferability beyond the baseline generation. It would be clearer if the paper specified whether the input is a clean image or a previously generated adversarial one.

3. Limited evaluation scope.
Only one dataset is used for experiments, which is insufficient to demonstrate the generalization of the proposed method. Transfer-based attacks typically require evaluations on multiple datasets (e.g., ImageNet, CIFAR, Tiny-ImageNet, etc.) to establish robustness and scalability.

4. Incomplete or missing baselines.
Several baselines are mentioned but not properly introduced or cited. For instance, ILPD is included in the tables but never described in the related work section. Furthermore, essential and widely recognized baselines such as AutoAttack are missing entirely. Without comparison to these strong and standard methods, the performance claims are not convincing.

5. Ambiguous relation to semantic transfer.
The core claim that low-rank features correspond to semantically meaningful information is intuitive but not empirically verified. The paper does not analyze whether the extracted low-rank subspaces indeed capture consistent semantics across architectures.

6. Writing quality and presentation issues.
The manuscript suffers from grammatical errors, inconsistent notation, and unclear figure explanations. Some equations are poorly formatted, and the structure of the methodology section is hard to follow. The presentation significantly affects readability and scientific clarity.

**Questions:**

see above

---

### Note · Authors · 2025-12-02

I have read and agree with the venue's withdrawal policy on behalf of myself and my co-authors.